# Exploring Cost-Effectiveness of the Comprehensive Geriatric Assessment in Geriatric Oncology: A Narrative Review

**DOI:** 10.3390/cancers14133235

**Published:** 2022-06-30

**Authors:** Sara Zuccarino, Fiammetta Monacelli, Rachele Antognoli, Alessio Nencioni, Fabio Monzani, Francesca Ferrè, Chiara Seghieri, Raffaele Antonelli Incalzi

**Affiliations:** 1Management and Health Laboratory, Institute of Management–Department Embeds, Sant’Anna School of Advanced Studies, 56127 Pisa, Italy; francesca.ferre@santannapisa.it (F.F.); chiara.seghieri@santannapisa.it (C.S.); 2Department of Internal Medicine and Medical Specialties (DIMI), Università di Genova, 16132 Genoa, Italy; fiammetta.monacelli@unige.it (F.M.); alessio.nencioni@unige.it (A.N.); 3IRCSS Ospedale Policlinico San Martino, 16132 Genoa, Italy; 4Geriatrics Unit, Department of Clinical & Experimental Medicine, Pisa University Hospital, 56126 Pisa, Italy; rachele.antognoli@gmail.com; 5Unit of Geriatrics, Department of Medicine, Campus Bio-Medico di Roma University, 00128 Rome, Italy

**Keywords:** CGA, older adults, geriatric oncology, cost-effectiveness

## Abstract

**Simple Summary:**

The Comprehensive Geriatric Assessment is a multidimensional and multidisciplinary evaluation designed for elderly patients with the goal of structuring tailored care and follow-up. Despite the known benefits of this approach, the Comprehensive Geriatric Assessment is not universally applied to elderly cancer patients due to economic and practical barriers. This narrative review aims to investigate the cost-effectiveness of the Comprehensive Geriatric Assessment adopted in geriatric oncology. The results revealed a lack of research on the topic, but recurrent cost-saving effects of this approach in geriatric oncology settings were highlighted—suggesting a positive cost-effectiveness ratio. Further structured research with comprehensive economic evaluations is needed to confirm these findings.

**Abstract:**

The Comprehensive Geriatric Assessment (CGA) and the corresponding geriatric interventions are beneficial for community-dwelling older persons in terms of reduced mortality, disability, institutionalisation and healthcare utilisation. However, the value of CGA in the management of older cancer patients both in terms of clinical outcomes and in cost-effectiveness remains to be fully established, and CGA is still far from being routinely implemented in geriatric oncology. This narrative review aims to analyse the available evidence on the cost-effectiveness of CGA adopted in geriatric oncology, identify the relevant parameters used in the literature and provide recommendations for future research. The review was conducted using the PubMed and Cochrane databases, covering published studies without selection by the publication year. The extracted data were categorised according to the study design, participants and measures of cost-effectiveness, and the results are summarised to state the levels of evidence. The review conforms to the SANRA guidelines for quality assessment. Twenty-nine studies out of the thirty-seven assessed for eligibility met the inclusion criteria. Although there is a large heterogeneity, the overall evidence is consistent with the measurable benefits of CGA in terms of reducing the in-hospital length of stay and treatment toxicity, leaning toward a positive cost-effectiveness of the interventions and supporting CGA implementation in geriatric oncology clinical practice. More research employing full economic evaluations is needed to confirm this evidence and should focus on CGA implications both from patient-centred and healthcare system perspectives.

## 1. Introduction

Conceived in the 1960s, the Comprehensive Geriatric Assessment (CGA) is “a multi-disciplinary process which includes assessment and management of assessed need” of older patients in relation to “medical, social and functional needs”, ending in “the development of an integrated/coordinated care plan to meet those needs” [1]. The CGA comprises different scales to evaluate each health dimension, and single geriatric screening tools are often adopted as alternatives to a full CGA, enabling a brief CGA [2,3]. The CGA has a strong prognostic value and is essential to identifying older adults at risk of a disability. Thus, it is considered a fundamental component of geriatric care [4].

In oncology, the assessment of a patient’s Performance Status (PS) is crucial for the treatment decision and prognosis estimation [5]. The PS predicts the treatment benefits and the risk of toxicity, and it is commonly required for patient enrolment in clinical trials [6]. Among the different PS scales, the Eastern Cooperative Oncology Group PS (ECOG PS) and the Karnofsky PS (KPS) scores are perhaps the most widely adopted tools [6,7]. However, ECOG PS and KPS are not sensitive enough to identify the functional limitations and essentially lose their prognostic value in older patients [8,9]. Instead, the CGA is more accurate in defining the actual degree of fitness of older patients. It allows to properly categorise patients as fit, vulnerable or frail, which, in turn, is supposed to help oncologists make more appropriate treatment decisions.

The CGA typically comes with recommendations/interventions that, by tackling the identified issues (e.g., depression, cognitive impairment, delirium, depression, malnutrition, sleep disorders, social issues, etc.), optimise a patient’s health status, help to regain an active treatment for some of the older patients [10,11,12] and may also increase a patient’s tolerance to anticancer treatments [13,14].

According to the International Society of Geriatric Oncology (SIOG), the CGA represents the gold standard for (a) defining a prognosis and the ability to withstand cancer treatments, (b) exploring the multiple aspects that define the complexity of frail older persons and (c) designing person-tailored interventions [2]. Similarly, the European Society for medical Oncology (ESMO) [15], the European Society of Surgical Oncology (ESSO) [16], the American Society of Clinical Oncology (ASCO) [17], the National Comprehensive Cancer Network (NCCN) [18], the American College of Surgeons (ACS) [19,20] and the Italian Society of Geriatrics and Gerontology (SIGG) [2] have also recommended the adoption of CGA in older cancer patients (see Table 1).

To what extent the CGA has an impact on oncologists’ decisions was studied by Chaïbi and colleagues [23], Girre and colleagues [24] and Marenco and colleagues [25]. These authors found that the CGA affected treatment decisions in a percentage of cases ranging between 21% and 49% [23,24,25,26]. Other studies have shown that incorporating the CGA in the management of older patients with cancer (i) improves clinical outcomes by helping select the most appropriate therapy [27], (ii) promotes the inclusion of patient preferences in the decision-making process [28,29,30], (iii) improves communication between oncologists and patients [28,29,30], (iv) reduces the risk of over- and undertreatment [31], (v) enhances treatment tolerance and completion [32] and (vi) predicts the frequency of hospitalisation and long-term care for older cancer survivors [33].

The CGA is also beneficial in terms of reducing the mortality, disability and institutionalisation of community-dwelling older people, allowing for the preservation of physical function, lower healthcare utilisation and reduced hospitalisations [34,35].

Although the majority of cancer patients are in the geriatric population [36,37,38], and despite the reported benefits of the CGA in the geriatric oncology setting, the CGA is not universally applied in older patients due to the lack of workforce (primarily of geriatricians); economic, logistical and practical barriers (e.g., time-consuming) [35] and because of the limited appreciation of the value of the CGA by oncologists. Gladman and colleagues (2016) referred to this issue as “a know-gap” as a way to express the uncertainty in adopting the CGA in specific settings, including clinical oncology [39]. 

Indeed, since most healthcare systems ultimately only have very limited resources available, showing that the CGA is cost-effective is necessary for it to become the standard of care [40].

Only a few trials have evaluated the costs and the impact of the CGA, resulting in tenuous evidence of its cost-effectiveness [41]. With some exceptions, the available studies suggest that the CGA and the corresponding geriatrics interventions are effective without raising the total cost of care [41]. The review by Fox and colleagues [42] showed that the cost of care in an acute geriatric unit was significantly lower than those of the usual care [42], and two studies concluded there was a reduction in the costs associated with the CGA for many of the hospital-based services analysed [43,44]. A recent RCT of geriatric co-management combined with an interdisciplinary transitional care intervention for frail older patients who had unplanned admissions to internal medicine services in Argentina showed a reduction in 30-day hospital readmissions and emergency department (ED) visits 6 months after discharge in the intervention arm [45]. The cost-effectiveness analysis of the Elder-Friendly Approaches to the Surgical Environment (EASE) Intervention for the emergency abdominal surgical care of older adults [46] conducted by Hofmeister and colleagues suggested that the EASE intervention was associated with a reduction in costs and no change in Quality-Adjusted Life Years (QALYs) [47].

On the other hand, Parker and colleagues [1] concluded that the available studies were lacking a “broader view”- meaning an analysis of the direct costs (i.e., staff and resources) but, also, of the subsequent costs (e.g., community health and social care costs) and of the costs for patients and the wider society [1]. 

In this review article, we aimed at investigating the cost-effectiveness of the CGA in a geriatric oncology setting. 

## 2. Methods

### 2.1. Search Strategy and Selection Criteria

#### 2.1.1. Data Sources

This review was based on a search of the PubMed and Cochrane databases for full papers and articles without restrictions based on the year of publication. The review search took place in between June 2021 and January 2022.

After duplicate exclusion, the article titles and abstracts were screened. The full texts were then screened and selected. The study characteristics and information were extracted from the selected papers. The SANRA scale for the quality assessment of narrative review articles was used to evaluate the review, adopting the revised version of the scale [48]. The revised SANRA scale is composed of six items rated from 0 (low standard) to 2 (high standard), with 1 as an intermediate score. The items cover: an explanation of the review’s importance (item 1) and statement of the aims (item 2) of the review, the description of the literature search (item 3), referencing (item 4), scientific reasoning (item 5) and a presentation of the relevant and appropriate endpoint data (item 6). Our review adopted an objective and systematic approach in the selection and analysis of the studies.

#### 2.1.2. Search Terms

-(comprehensive geriatric assessment OR (comprehensive geriatric assessment AND cancer) OR (comprehensive geriatric assessment AND oncology) OR (geriatric assessment AND cancer) OR (geriatric assessment AND oncology) OR (geriatric evaluation management) OR (geriatric evaluation management AND cancer) OR (geriatric evaluation management AND oncology) OR (geriatric co-management)) OR (geriatric co-management AND cancer) OR (geriatric co-management AND oncology) OR (geriatric comanagement) OR (geriatric comanagement AND cancer) OR (geriatric comanagement AND oncology) OR (geriatric intervention AND cancer) OR (geriatric intervention AND oncology)) AND (cost-effectiveness OR cost OR expenditure OR cost-utility OR utility-analysis);-(([comprehensive geriatric assessment] AND [cancer]) OR [geriatric comanagement AND cancer] OR [geriatric comanagement AND oncology]) (([comprehensive geriatric assessment AND oncology] OR [geriatric assessment AND cancer] OR [geriatric assessment AND oncology] OR [geriatric evaluation management AND cancer] OR [geriatric evaluation management AND oncology] OR [geriatric co-management AND cancer] OR [geriatric co-management AND oncology] OR [geriatric intervention AND cancer] OR [geriatric intervention AND oncology]) AND (length of stay OR LOS) OR (readmission) OR (falls) OR (complications) OR (emergency department);-((comprehensive geriatric assessment) OR (geriatric assessment) OR (geriatric comanagement) OR (geriatric co-management) OR (geriatric evaluation management) OR (geriatric intervention)) AND toxicity).

#### 2.1.3. Study Eligibility Criteria

To select the studies, as the eligibility criteria, we set (a) the focus on older cancer patients (60 years or older), both sexes, with a diagnosis of cancer and cared for in geriatric oncology settings, or oncology or surgery, or included in heterogeneous geriatric populations in a medical or surgical setting (with a consistent rate of older patients with cancer in the study population); (b) with the implementation of a full CGA or a brief CGA with at least one CGA tool and (c) the recurrence of the cost measures related to the adoption of the CGA (i.e., costs and/or resources required for the management of older patients) and/or the presence of measures for the effectiveness of the CGA intervention and/or cost-sensitive outcomes (i.e., outcomes with cost-effectiveness implications).

Moreover, such studies were retrospective, prospective cohorts, observational or interventional in nature, with at least 35 included patients. 

The records were screened for inclusion based on predefined criteria. The papers were excluded if they concerned:Editorials, protocols, score creation studies, ongoing registered trials or completed trials without available results;Studies without a specific focus on older adults (i.e., age < 60 years or no data about old age participants);Studies without a cancer population or without a reported percentage of cancer patients in the study cohort;Studies without measures of the cost-effectiveness or cost-sensitive measures;Nursing home patients/patients receiving home care;Studies enrolling less than 35 patients.

#### 2.1.4. Analysis of Studies

The studied identified were analysed according to the number and characteristics of the participants (i.e., age, cancer site, stage, therapy and frailty); the study design; the type of CGA and geriatric interventions; the main effects of the interventions and the reported measures of effectiveness suitable for cost-effectiveness or the cost-effectiveness measures when available.

For each study, the (cost-)effectiveness measures reported were classified as (i) cost-sensitive measures (measures of patient health conditions leading to cost-increasing or cost-decreasing effects) and/or (ii) measures of the effectiveness of the CGA intervention (with an estimation of the costs in two studies). The cost-effectiveness propensity of each measure was assessed—that is, the propensity of that measure to sound cost-effective in light of the results obtained in a specific study. Each measure was classified as “positive” when leading to implications that improve the patient health status with a cost-saving or cost-decreasing effect. Conversely, the term “negative” was given to measures that increase the costs. The term “neutral” depicts the measures that present neither positive or negative implications with respect to the cost trend or resource exploitation. 

The reason for a qualitative assessment of the propensity lies in the lack of cost-effective measures or cost calculations for the majority of the studies, preventing a true quantitative cost-effective evaluation.

## 3. Results

### 3.1. Identification of Relevant Studies

The literature search yielded 8613 potentially relevant papers, with 142 duplicates that were removed. After the removal of the duplicate records, protocols, ongoing studies—as for two conference abstracts [49,50,51], editorials and records without relevance to the research question, either for their research focus or because they utilised the CGA in non-oncological settings—from abstract and full-text screening, a total of 37 records were assessed for eligibility in the review, including 5 records retrieved during a manual search (Figure 1).

A large number (8613) of articles were identified from the database searches, and 16 additional articles were identified by exploring their bibliographies and by a manual search. 

Out of the 37 eligible studies, two studies were not included due to the characteristics of interventions [52,53], two studies were not included because it was not possible to identify the percentage of cancer patients [45,54], two studies were excluded because the number of cancer patients included was very low [46,55] and another two studies were excluded due to nonexplicit criteria for patient inclusion concerning age [56] or the inclusion of patients of all ages [57]. 

In the end, 29 studies were identified for the review: 24 retrieved from the databases and five from citation searching or the manual search. Twenty-six full-text studies and three conference abstracts showed the results of completed trials [58,59,60]. All extracted data from the selected articles were categorised according to the study design, study participants and measures of cost-effectiveness.

The literature we found with valuable measures for the cost-effectiveness of the CGA implemented in older cancer patients consisted of 29 studies. 

Among the identified studies, there were 15 RCTs [58,59,61,62,63,64,65,66,67,68,69,70,71,72,73], 6 observational cohort studies [74,75,76,77,78,79], 3 before-and-after studies [80,81,82], 2 pilot studies [60,83], 1 descriptive comparison study [84], 1 prospective observational study [85] and 1 secondary analysis study (of RCT) [86].

Among the studies, 13 were carried out in surgical settings (Table 2) and 16 in medical settings (Table 3). In Table 2 and Table 3, we list the details of the studies retrieved.

The cancer site considered is colorectal cancer (CRC) in 10 studies [59,60,64,67,75,77,78,79,80,81]; CRC or esophagogastric cancer in 1 study [82]; gastrointestinal cancer (GI) in 1 study [72]; GI, gastro–urinary (GU) or breast cancer in another study [73]; GI or lung cancer in another study [70]; GI or other sites in another study [74]; GU, bladder, kidney or NSCLC in another study [69]; head and neck, lung, upper GI or CRC in another study [71]; NSCLC in another study [66]; prostate, lung, haematologic, GI, head/neck, bladder, renal and other sites in another study [86]; lymphoma, leukaemia or multiple myeloma in another study [65]; solid cancers in 4 studies [61,63,83,85]; solid cancers or lymphoma in another 4 studies [58,62,68,84] and different malignant neoplasms in another study [76].

The mean ages of the patients included in the identified studies varied between 72 and 83.7, years and the median ages ranged from 71 to 83 years.

The included studies published using data from Europe (12 studies [63,66,67,68,69,71,75,77,78,79,80,85]), followed by the USA (9 studies [28,61,62,70,72,83,84,86,90]), Hong Kong (2 studies [59,60]), the UK (2 studies [74,82]), Australia (1 study [58]), Canada (1 study [73]), Japan (1 study [64]) and Singapore (1 study [81]). Four articles were published between 2005 and 2014 [60,63,68,86], ten between 2015 and 2018 [59,64,66,73,74,77,79,82,83,84] and fifteen between 2019 and 2022 [28,58,61,62,67,69,70,71,72,75,76,78,80,81,85].

### 3.2. Studies Assessing the Effectiveness and Costs of CGA

#### Surgical and Medical Setting

Two studies out of the selected articles included measures of effectiveness of the CGA and estimated the direct costs of the intervention compared to the usual care. Even though the incremental cost-effectiveness ratio (ICER) was not performed, the cost-effectiveness of the interventions results were positive, since the CGA improved the outcomes at a lower cost when compared to the control group - by adopting the same approach already used by Eamer and colleagues [89] or improved the outcomes without raising the costs.

The first was a nonrandomised prospective before-and-after study conducted by Koh and colleagues that recruited 81 patients (≥70-years) with resectable colorectal cancer (CRC) scheduled for elective colectomy [81]. The geriatric intervention consisted of a standardised prehabilitation program (Programme for Enhanced Elderly Recovery at Sengkang General Hospital—PEERS). The program included the CGA, nutrition supplementation, resistance training, optimisation of the cardiac risk for operation and early evaluation of the patient home to ensure the residence was equipped to receive the patient after surgery. In addition to good results in the QoL for the PEERS group compared with the patients from the control arm (who were treated according to a standard-of-care approach), the average duration of hospitalisation in the PEERS group was 6.8 days shorter—after adjusting for surgical approach and complications—with an average USD 11,838.80 savings per patient [81]. 

The second, by Rao and co-authors, analysed from a health economic perspective a RCT that was conducted in 11 medical centres [86]. In this trial, 99 frail older patients (≥65 years)—who were hospitalised either in a medical or in surgical ward with a miscellaneous group of cancer diagnoses (solid and haematological malignancies)—were randomised to receive geriatric care or standard care. In the experimental arm, the CGA and the corresponding interventions were implemented by a physician, a nurse practitioner and a social worker. Although there was no effect of the CGA-driven interventions on the mortality and on overall QoL (36-Item Short-Form general health survey—SF-36 score), the inpatients who were treated according to a geriatric approach exhibited better mental health, less bodily pain and lower emotional limitation on the SF-36 scale than the usual inpatients at discharge. Overall, there was no significant difference between patients who received CGA-driven interventions vs. patients who were managed according to the standard of care in terms of the total hospital costs after one year (USD 47,300 vs. USD 45,500, respectively). Similarly, the total geriatric outpatient costs were also not significantly impacted by the intervention (USD 44,700 vs. USD 48,100 for patients managed through the CGA and patients managed according to the standard of care, respectively), as well as LoS. In this study, the costs that were taken into account included those of inpatient, outpatient and long-term care provided by Veterans Affairs medical centres, whereas the costs of inpatient and outpatient care in other facilities, as well as care in private nursing homes, were not included [86].

### 3.3. Studies Reporting the Outcomes of CGA Interventions Other Than Treatment Toxicity

We identified 17 articles reporting the outcomes of interventions—often also used to rate the cost-effectiveness—in relation to CGA adoption in older cancer patients who were treated either in a surgical or in a medical setting. 

#### 3.3.1. Surgical Setting

Several studies of the CGA in older patients undergoing oncological surgery have been identified.

In the study conducted by Janssen and co-authors, a multimodal prehabilitation program—tailored to reduce the incidence of delirium and other adverse events (AEs) in older patients undergoing elective major abdominal surgery—had no effect on the Length of Stay (LoS), readmissions, unplanned ICU admissions, institutionalisation and postoperative complications but successfully reduced the incidences of delirium [80]. In the study conducted by Tarazona-Santabalbina, CGA-based interventions resulted in a lower incidence rate of delirium and other geriatric syndromes in the intervention group admitted for elective CRC surgery without a significant effect on readmissions, even if these patients had significantly poorer functional conditions, a higher prevalence of dementia and heart failure and a higher comorbidity burden at the baseline [75], contributing to serious complications that were more frequent in this group. A multicentre prospective RCT conducted by Hempenius and colleagues evaluated the effect of the CGA and CGA-driven interventions on the incidences of postoperative delirium (PoD) in older cancer patients (≥65 years) undergoing elective surgery for solid tumours [63]. The CGA intervention consisted of a preoperative geriatric consultation, an individual treatment plan targeted at the risk factors for delirium, daily visits by a geriatric nurse during the hospital stay and advice on how to manage the ensuing medical problems. The intervention failed in the main purpose, since there was no significant difference between the incidence of PoD in the intervention group and the usual care group. Similarly, there was no effect of the CGA on the postoperative complications, mortality or care dependency post-discharge. However, this study did find a positive effect of the CGA on bodily pain (SF-36 domain), with no increase in LoS [63].

In the retrospective study that was conducted by Indrakusuma and colleagues, the group undergoing a geriatric preoperative assessment (Dutch acronym: DOG)—consisting of a CGA and geriatric interventions—presented a higher prevalence of a history of delirium than the controls, but the tailored intervention resulted in a lower PoD for DOG patients compared to the controls [79]. Moreover, being at a higher risk than the controls, DOG patients had comparable postoperative general/surgical and medical complications, with no significantly shorter LoS and similar outcomes in mortality [79].

Ommundsen and colleagues showed that a preoperative geriatric assessment reduced the total number of grade I–V complications without increasing the LoS and reoperation rate in frail patients who received elective CRC surgery [64]. No statistically significant differences were found between the intervention group and the control group for reoperations or readmissions [64].

In the study by Shipway and colleagues, 132 patients underwent preoperative CGA and profited from geriatric medical care during hospitalisation [82]. Among the patients undergoing CGA, 36% of the patients who received a preoperative CGA were deemed noneligible for surgery. Overall, this study found that the CGA and the involvement of a geriatrician in patient care resulted in a LoS reduction of 3.1 days among the patients older than 60 years. The effect of the CGA and geriatric intervention was also found in patients (older than 60 years) who received emergency surgery, with a mean LoS reduction of 4.4 days. Conversely, in patients admitted for elective GI cancer surgery, no statistically significant LoS reduction was observed, although the authors reported a trend towards a reduction with advancing age (particularly in patients ≥75 years who had a 5.2-day reduction in LoS) [82].

In the study of Ho and colleagues on older patients with colorectal cancer randomised to conventional surgical care or enhanced geriatric input, the median LoS was significantly shorter, and the postoperative complications were significantly lower in the intervention group when compared to the control [59].

Similarly, in a pilot study focused on enhanced geriatric input in the management of older patients undergoing CRC surgery conducted by Mak and co-authors, the patients who received geriatric input showed a shorter mean LoS and lower 30-day morbidity when compared with the controls [60].

In the cohort study carried out by Shahrokni and colleagues on older patients undergoing cancer-related surgical treatment, the intervention patients—receiving geriatric co-management—were older than those who received care managed by the surgery service only, but there were no differences in frailty measured by the Memorial Sloan Kettering Frailty Index (MSK-FI) [76]. Patients receiving co-management had longer operative times and longer LoS than the control group [76]. On the other hand, a higher proportion of patients in the geriatric co-management group received inpatient supportive care services, including physical therapy, occupational therapy, speech and swallow rehabilitation and nutrition services, and they had significantly lower 90-day postoperative mortality. The adverse surgical outcomes within 30 days of surgical treatment did not differ across the groups [76].

In the study conducted by Souwer and co-authors on older patients that underwent elective surgery for stage I–III CRC, the patients who benefited from a comprehensive multidisciplinary prehabilitation and rehabilitation care program showed a lower one-year overall mortality, a significant reduction in cardiac and severe complications and in the number of patients with a prolonged LoS [77].

The cohort study carried out by van der Vlies and colleagues on frail older patients with CRC undergoing surgery resulted in comparable postoperative outcomes for frail patients compared to non-frail patients by means of a CGA intervention; they were even found at an increased risk for a worse OS [78].

In the RCT conducted by Nipp and co-authors on older patients with GI cancer undergoing surgery, the perioperative geriatric intervention (PERI-OP) did not have a significant impact on ICU use, hospital readmissions or complications. However, in the PP analysis, the subgroup who received PERI-OP as planned experienced significantly shorter postoperative hospital LoS [72].

#### 3.3.2. Medical Setting

Concerning the studies of the CGA and GCA-driven interventions in older patients undergoing medical treatment, again, several examples of this approach are available in the literature, and we identified studies reporting measures sensitive to implications for the cost-effectiveness.

In the study conducted by Soo and co-authors evaluating older cancer patients who received systemic anticancer therapy, an integrated oncogeriatric approach led to improvements in health-related QoL, treatment discontinuation and a reduction in unplanned hospital admissions and ED visits [58].

DuMontier and colleagues found that, in prefrail and frail older patients with haematologic malignancies, an embedded geriatric consultation did not improve the acute care utilisation (not significantly reduced LoS, hospitalisations and ED visits), but it significantly increased the likelihood of having End-of-Life (EoL) goals of care discussed without an increase in acute care [65].

In the pilot RCT of a transdisciplinary intervention integrating geriatric and palliative care with oncology care carried out by Nipp and colleagues, consequently, in visits with a geriatrician and CGA intervention patients—with incurable GI or lung cancer—had less decrease in QoL decrement, a reduced number of moderate/severe symptoms and improved confidence in communication compared to the usual care [70].

In the RCT of a tailored follow-up intervention on CGA in older frail patients with cancer conducted by Ørum and colleagues, even without a significant rate—a lower percentage of patients in the intervention group—were admitted to hospital during the study (47% vs. 55% of controls), while no differences in ability to complete the treatment, ADLs or the physical performance were found [71].

In the RCT carried out by Puts and co-authors [73], the CGA-based intervention on older cancer patients induced a slight benefit in QoL for the intervention patients, but in the secondary analysis conducted by Sattar and colleagues, the results of EQ-5D-3L at all timepoints presented no statistically significant differences between the two groups [88]. Sattar and colleagues also found no statistically significant differences between the intervention and control group in the number of ED and family physician visits, even with slightly lower rates in the intervention group [88].

### 3.4. Studies Reporting Outcomes of CGA Interventions: Treatment Toxicity and Other Complications

#### Medical Setting

Ten of the studies we identified investigated the impact of CGA in terms of the treatment toxicity, almost all resulting in a toxicity reduction, a good cost-sensitive outcome in representative cohorts of older cancer patients. 

Li and colleagues conducted a RCT on older patients (≥65 years) with a solid malignant neoplasm (GI 33.4%, breast 22.5%, lung 16.0%, GU 15.0%, gynaecologic 8.9% and other cancer types 4.1%) who were starting a new chemotherapy regimen and completed a geriatric assessment [61]. Six hundred and five patients were enrolled in this trial. According to this study, a CGA-driven intervention implemented by a multidisciplinary team (oncologist, nurse practitioner, social worker, physical/occupation therapist, nutritionist and pharmacist who reviewed the CGA results and implemented interventions based on prespecified thresholds) significantly reduced the grade III/higher chemotherapy-emergent AEs. No significant differences were observed in the average LoS, unplanned readmissions and hospitalisations, ED visits and overall survival (OS), as well as in chemotherapy dose modifications or discontinuations [61].

Similarly, Mohile and colleagues performed a randomised controlled trial enrolling 718 patients (≥70 years) with an incurable solid cancer or lymphoma, at least one impaired CGA domain and who were starting a new treatment regimen [62]. The intervention that was evaluated was CGA-coupled with the geriatric assessment-guided management recommendations provided to the community oncologists. This trial found that the CGA and GCA-driven recommendations largely reduced the serious chemotherapy-emergent (grade III–V) ADEs and falls in patients with advanced cancer and aging-related conditions [62].

Choukroun and colleagues conducted a single-centre prospective study among 51 older outpatients with cancer (≥75 years), showing that CGA combined with a pharmacist consultation was effective at detecting and contrasting the use of potentially inappropriate medications (PIMs) [85].

Kalsi and colleagues evaluated the impact of CGA-driven interventions on chemotherapy toxicity and tolerance in older patients (≥70 years) with cancer undergoing chemotherapy [74]. The authors found geriatrician-led CGA interventions to be associated with improved chemotherapy tolerance with a reduced rate of (grade III/higher) toxicity (even if not significantly) after adjusting for age, comorbidity, metastatic disease and initial dose reductions [74]. 

In the study conducted by Corre and co-authors on older patients with advanced Non-Small-Cell Lung Cancer (NSCLC), treatment allocation based on the CGA slightly reduced the treatment toxicity [66].

In a RCT with 142 patients, Lund and colleagues showed that geriatric interventions increased the rate of completion of adjuvant chemotherapy and QoL, reducing the toxicity, for frail older patients receiving chemotherapy for CRC [67]. Furthermore, more patients from the CGA arm completed the scheduled chemotherapy compared with the controls [67].

In the study conducted by Ramsdale and co-authors on 40 older patients with advanced solid cancer or lymphoma, more PP concerns were brought up and addressed in the intervention CGA group [84].

In the two-year RCT on nutritional advice conducted by Bourdel-Marchasson and colleagues on older patients treated with chemotherapy for carcinomas and lymphomas at risk of malnutrition, the diet counselling was efficient in increasing the dietary intake but had no beneficial effect on one-/two-year mortality; chemotherapy management (dosage, changes and arrest) and hospitalisations, even if the intervention group experienced non-significantly fewer hospitalisations [68]. There were more usual care patients with grade III to IV infections than in the intervention group, but the robustness analysis did not confirm the difference in the incidence of severe infections [68]. 

The results of the RCT carried on by Nadaraja and colleagues on 96 patients shown oncologic treatment allocation for frail older cancer patients based on G8 screening followed by CGA had no impact on treatment completion, OS or median progression-free survival (PFS) but resulted in a borderline significant lower incidence of grade III to IV toxicity in the intervention group compared with the control group [69].

In the pilot study conducted by Magnuson and co-authors on 71 older patients with advanced (stage III or IV) solid tumour malignancy, only 35.4% of the CGA recommendation was implemented by the primary oncologist, and the incidence of grade III–V chemotherapy toxicity did not differ between the intervention and control groups, as well as the prevalence of hospitalisation, dose reductions, dose delays and early treatment discontinuation [83].

The studies identified are heterogenous for what concerns the setting, the sample size, the characteristics of the patients (for example, different types of malignancies and presenting with frailty or geriatric syndromes) and the aim of the CGA [91]. 

The parameters specifically adopted to rate the cost-effectiveness retrieved in these studies are scarce; only two studies reported effectiveness measures and costs, allowing to compare the outcomes and change in costs [81,86]. These studies focused on resource utilisation, institutional care costs, costs for readmissions and costs of direct health service uses.

Twenty-three studies included the measures of effectiveness of CGA interventions in terms of clinical outcomes and quality of life, such as Quality-Adjusted Life Years (QALYs) or proxy outcomes, the most recurrent being hospital readmissions and reoperations, LoS, unplanned hospitalisations and ICU or ED admissions (grey rows in Table 2 and Table 3) [28,59,60,61,63,64,66,67,68,70,71,72,73,75,77,78,79,80,81,82,83,86,90]. All these measures have been frequently adopted in the literature to rate the cost-effectiveness—in particular, LoS as a proxy measure for resource use [92,93], here interpreted as cost-driven or cost-saving measures, a proxy for the propensity of cost-effectiveness of the specific CGA intervention, since the studies lack of actual cost estimations, with the exemption of the studies by Rao and colleagues [86] and Koh and co-authors [81]. 

Furthermore, almost all studies have reported cost-sensitive measures—measures of patient health conditions implying cost-increasing or cost-decreasing effects, even if not estimated in terms of costs, such as postoperative complications, toxicity, PP, PIM, therapy completion, CT tolerability and falls (white rows in Table 2 and Table 3). In almost all studies, at least one dimension was positively impacted by the adoption of CGA in older patients. The LoS and severity of ADEs are the most reported measures, with a plausible good cost–benefit rate in the majority of studies (Table 4).

## 4. Discussion

The limited number of studies retrieved in our review indicates the lack of research on the topic of cost-effectiveness implications in CGA interventions. This result is consistent with the scarcity of combined medical and economic evaluations of older patients’ care [94].

Despite the limited number and the large heterogeneity among the studies identified in our review, the overall evidence rate is in favour of a measurable benefit from the CGA in the management of older patients with cancer in terms of reduced LoS (or at least stable LoS) and treatment toxicity and in improved clinical outcomes both in the medical and surgical settings (see Table 2 and Table 3).

These results contribute to the evidence for an “investment effect” of the CGA. This concept—originated in the 1980s—was recalled by Wieland in his review of the CGA cost-effectiveness, meaning the investment of resources in the patient, not in the entire cohort of patients, providing more appropriate services wherein the costs of more appropriate care are offset by the less use of expensive institutional services [41].

Several studies showed that the CGA improves the profiling of older patients and, as a result, the tailoring of treatments. A validated prediction model designed by Hurria and colleagues was proven to independently predict the risk of treatment toxicity [9,95], as well as Extermann and colleagues, who elaborated and validated the Chemotherapy Risk Assessment Scale for High-Age Patients (CRASH), allowing for toxicity risk-stratification across a wide range of chemotherapies [96]. Ultimately, the CGA could well also be cost-effective thanks to its ability to select more patients for the best supportive care (instead of active treatment), preventing patients from experiencing severe toxicity and a rapidly worsening quality of life but also saving the costs of expensive anticancer medications and of the hospitalisations and treatments required for chemotherapy AEs (i.e., antibiotics, recombinant hematopoietic growth factors, etc.) [66]. Orienting the therapeutic choices, CGA instruments can also offer a valid contribution in avoiding overtreatments that lead to a worsening of the quality of life and avoidable costs. In Table 5, we report the costs of neutropenia, thrombocytopenia and anaemia, the most frequent haematological complications of cancer treatments [97].

Seven of the identified studies in our review showed a positive effect of the CGA on treatment tolerance and toxicity. In the study conducted by Kalsi and colleagues [74], a geriatrician-delivered CGA was associated with better outcomes for older people undergoing chemotherapy (higher frequency of treatment completion and lower frequency in treatment modifications), with a lower chemotherapy toxicity, even if not significantly. Mohile and co-authors [62], Choukroun and colleagues [85], Lund and co-authors [67], Li and co-authors [61], Nadaraja and colleagues [69] and Corre and co-authors [66] showed that the adoption of the CGA resulted in a lower treatment toxicity. These results imply large cost reductions in the face of improvements in clinical outcomes, suggesting that interventions are cost-effective.

Table 5 shows an estimate of the costs that hospitals incur in for the management of geriatric cancer patients, including ADEs (these costs are extrapolated from the literature).

Among the evidence collected in our review, the CGA also appears to be an effective approach to symptom management and in assisting the management of pain and the emotional and mental health in older cancer patients, resulting in sustained improvement in the quality of life with no increase in costs [63,86]. These results were obtained in a medical [86] and in a surgical setting [63]. In the study conducted in a medical setting [86], the overall costs for institutional care were calculated and resulted equally in between the intervention and control group. In the other study [63], the LoS was similar between the intervention and control group, suggesting that a positive effect in the quality-of-life outcomes was obtained in a cost-effective way.

This review had some limitations. The studies included frequently enrolled small numbers of patients with different study designs. RCTs are present [58,59,61,62,63,64,65,66,67,68,69,70,71,72,73], together with cohort studies [44,102], before-and-after studies [80,81,82], pilot studies [60,83], a descriptive comparison study [84], a prospective observational study [85] and a secondary analysis study (of RCT) [86]. In addition, the CGA and CGA-driven interventions are not standardised. The cancer sites and outcomes reported are heterogenous, as well as genre balance and mean age of the study populations, even if the focus of all the studies was on older patients. Furthermore, it is difficult to analytically assess the cost-effectiveness of interventions because of the lack of studies implementing full economic evaluations and the heterogeneity among the studies in terms of the outcomes and cost estimations. For this reason, our analysis shifted only for qualitative arguments.

Currently, the lack of research on the CGA in oncology and on the cost-effectiveness of CGA-driven interventions does not allow to evaluate the cost-effectiveness of all the clinical benefits the CGA provides, including, for instance, the reduced risk of the institutionalisation of cancer patients [44,102] and the improved appropriateness of care [103].

More research is needed on the cost-effectiveness of the CGA in geriatric oncology, as well as the adoption of standard measures for this purpose. 

Moreover, to enable the adoption of CGA in oncology, a solid interprofessional collaboration and a careful choice of the right instruments are crucial [104,105]. Mckenzie and colleagues proposed to leverage information technologies to reduce the CGA implementation costs and to enable implementation of the CGA without the need for a dedicated geriatric oncology team/service [35]. This can be a valid choice, but the role of geriatricians in interpreting the CGA and in prescribing the resulting intervention remains crucial [106]. To pursue the minimisation of the costs, a good approach is integrating the geriatrician-led services required into existing structures (e.g., internal liaison and geriatric day clinic), promoting inter-speciality cross-fertilisation [35].

## 5. Conclusions

Our review highlights the lack of research on the topic of the cost-effectiveness implications of CGA interventions. Altogether, the results of our review support an “investment effect” of CGA in oncology. Despite overall not being tailored to rate the cost-effectiveness by design, the available evidence suggests that the CGA provides measurable benefits in older cancer patients with cost-savings effects, such as reductions in LoS—or stability of LoS in the face of improved clinical outcomes—and decrease in ADEs, leaning toward a positive cost-effectiveness of the CGA in geriatric oncology. However, more research employing full economic evaluations is needed to confirm this evidence. Further, dedicated studies are needed to optimise the CGA approach for different settings and to tailor CGA instruments to the available human and professional resources.

## Figures and Tables

**Figure 1 cancers-14-03235-f001:**
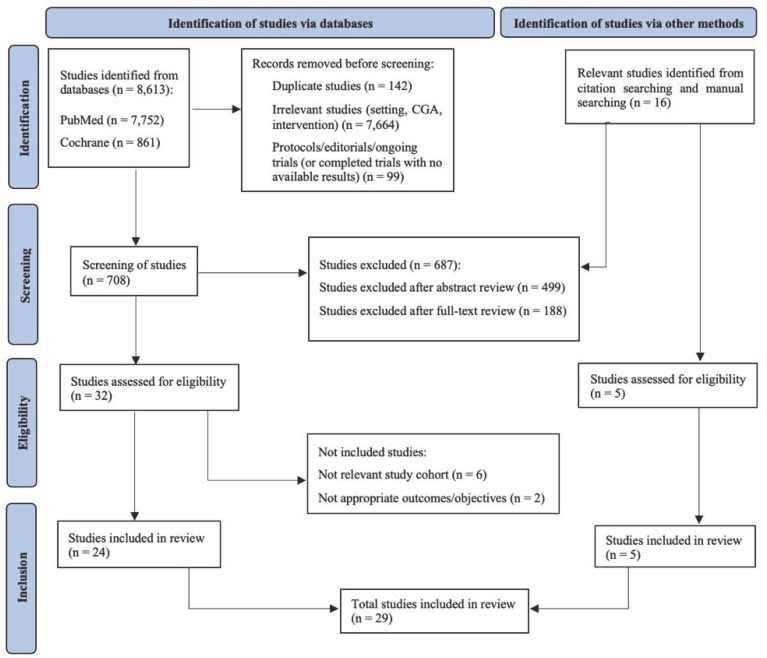
Review diagram.

**Table 1 cancers-14-03235-t001:** Recommendations in the CGA by scientific associations.

Proponents	Target Population	Recommendations
The International Society of Geriatric Oncology (SIOG) [21,22]	Cancer patients aged 70 or older	- adoption of CGA with further interventions and follow-up - use of screening tools to identify those patients who need CGA and multidisciplinary approach, when facing a busy clinical practice - the preferred screening tool may depend on the clinical situation
The European Society for Medical Oncology (ESMO) [15]	Patients aged 70 or older with diffuse large B-cell lymphoma (DLBCL)	- CGA is recommended to aid patient’s categorisation into fit, vulnerable and terminally ill patients
The American Society of Clinical Oncology (ASCO) [17]	Cancer patients aged 65 or older receiving chemotherapy	- CGA results should lay the basis of an integrated and individualised care plan - CGA results should inform cancer management - clinicians should consider CGA results when recommending chemotherapy - the information from CGA should be provided to patients and caregivers to guide treatment decision making
The National Comprehensive Cancer Network (NCCN), U.S.A. [18]	Older cancer patients	- implementation of pre-treatment evaluation using CGA when there are concerns about the patient’s ability to tolerate treatment or when issues are identified by a geriatric screening tool - assessment using a geriatric screening tool only, when there are no concerns regarding a patient’s ability to tolerate therapy - CGA should inform targeted interventions and a coordinated plan for cancer treatment
The Italian Society of Geriatrics and Gerontology (SIGG) [2]	Cancer patients aged 65 or older	- CGA should be performed in all patients with cancer aged 65 or older - as a second choice (whenever resources are limited), adoption of a ‘two-step approach’ (screening to select patients who need CGA) - a closer interaction between geriatricians and oncologists (or haematologists) should take place to optimise the approach to cancer patients - CGA should be performed by a trained geriatrician with the ability to detect and treat impairments in the different domains, possibly calling into play additional health professionals as needed
European Society of Surgical Oncology (ESSO) [16]	Patients aged 70 or older with rectal cancer	- CGA and multidisciplinary evaluation to identify the main predictors of frailty and postoperative complications such as functional status, nutritional status and comorbidities
American College of Surgeons (ACS) (and the American Geriatrics Society—AGS) [19,20]	Older cancer patients undergoing oncological surgery	- a preoperative frailty assessment is recommended for all older patients who are candidates to an oncological surgical procedure - risk assessment for older patients; management of geriatric domains in the perioperative period, postoperative period and after discharge

**Table 2 cancers-14-03235-t002:** Studies conducted in surgical settings, main features and results.

Authors	Participants	Type of Study	CGA and Geriatric Interventions	Effectiveness Measures Suitable for Cost-Effectiveness	Main Effect of CGA and Geriatric Interventions	Cost-Effectiveness Propensity
Nipp et al., 2022 [72]	98 patients (per-protocol) ≥65 years Type of cancer: GI [pancreatic, gastric/oesophageal, CRC, hepatobiliary cancer] - patients undergoing curative/palliative resections - 30 intervecntion patients	RCT	Random assignment to PERI-OP or usual care. Patients assigned to PERI-OP met with a geriatrician preoperatively and postoperatively. Geriatricians communicated findings to surgical/oncology teams or (after surgery) to inpatient team.	Postoperative LoS Postoperative ICU use 90-day hospital readmissions	In per-protocol analyses, PERI-OP patients had shorter postoperative LoS (5.90 vs. 8.21 days, *p* = 0.024), with non-significantly lower rates of postoperative ICU use and 90-day readmissions.	Positive Neutral Neutral
				Complications	In per-protocol analyses, differences in CD complication rates between PERI-OP and usual care group (6.7% vs. 20.6%, *p* = 0.137) were non-significant.	Neutral
Koh et al., 2021 [81]	81 patients ≥70 years Type of cancer: CRC (elective surgery) - 58 patients in intervention group - sequential comparison with earlier cohort (non-PEERS) of 23 CRC patients of a similar age who underwent colorectal resections and managed by the same group of surgeons Median age PEERS: 78.5 years; non-PEERS: 77 years. Female PEERS: 43%; non-PEERS: 48%.	Before-and-after	A structured multidisciplinary prehabilitation program prior to surgery: PEERS. The program included CGA, nutrition supplementation, resistance training, optimisation of cardiac risk for operation, optimisation of the discharge process to avoid institutionalisation. The control group did not benefit of PEERS program.	QoL LoS	PEERS group had significant improvement in median EQ-5D (0.70 pre-surgery to 0.80 6-months’ post-surgery, *p* = 0.01). After multivariate analysis, average LoS in PEERS group was 6.8 days shorter (*p* = 0.018) after adjusting for modality of surgery and complications, representing a cost saving of USD 11,838.80 per patient.	Positive
				Surgical complications Anthropometric and functional characteristics 30-days’ morbidity rate	Rate of CD grade 3+ complications were similar between groups. No significant improvement of anthropometric and functional characteristics before and after PEERS. Both groups had similar 30-days’ morbidity rates (8.6% vs. 17.4%, *p* = 0.26).	Neutral
Shahrokni et al., 2020 [76]	1892 patients ≥75 years Mean age: 81.5 years Female: 50% Type of cancer: different malignant neoplasms (and different surgical procedures). - 1020 patients in intervention group	Retrospective	Geriatric co-management of care with pre-operative (electronic Rapid Fitness Assessment) and postoperative evaluations.	LoS	Patients in the geriatric co-management group were older compared with surgical service group. The intervention group had longer operative time and longer LoS.	Negative
				Adverse surgical events Discharge destination	CD adverse surgical outcomes within 30 days of surgical treatment did not differ between groups. A higher proportion of patients in the geriatric co-management group were discharged home with home supportive services (18.0% vs. 13.6%, *p* < 0.001). In fully adjusted model, geriatric co-management was significantly associated with reduced 90-day mortality.	Neutral Positive
van der Vlies et al., 2020 [78]	433 patients ≥70 years Median age: 80 years (MDT patients), 75 years (non-MDT patients). Female: 41% Type of cancer: (stadium I-IV) CRC (elective curative surgery) - 127 frail patients in intervention arm (considered frail by clinical judgment or with G8 and 6-CIT)	Retrospective	Intervention: extended preoperative CGA. MDT estimated the risk of a surgical procedure and when patients were considered eligible for surgery, a prehabilitation program was initiated based on comorbidity and frailty characteristics. Control group: no preoperative MDT approach.	LoS Readmission rate Unplanned ICU admission	Readmission rates were similar between groups and most frequently caused by an infectious complication. No significant results were found for LoS and unplanned ICU admissions.	Neutral
				Severity of postoperative complications Discharge destination	Despite at increased risk, MDT patients did not suffer more postoperative CD III-V complications than non-MDT patients (14.9% vs. 12.4%; *p* = 0.48). Control group was discharged independently at home more often than MDT patients.	Neutral
Janssen et al., 2019 [80]	627 patients ≥70 years Mean age: 76 years Female: 36% Type of cancer: CRC (79.7%) [or patients with abdominal aortic aneurysm (20.3%)] (elective abdominal surgery) - 267 patients in the intervention group (CRC 73.8%, abdominal aortic aneurysm 26.2%)	Before-and-after	Prehabilitation group received interventions to improve physical health, nutritional status, factors of frailty and preoperative anaemia prior to surgery. During the outpatient visit, a nurse practitioner and a physiotherapist re-evaluated patient global health, fitness and frailty. The control group was not pre-habilitated.	In-hospital LoS Hospital readmission rate Unplanned ICU admission ICU LoS	The prehabilitation group had a higher burden of comorbidities and was more physically and visually impaired at baseline. No effect of prehabilitation on LoS, readmissions, unplanned ICU admissions and LoS in ICU.	Neutral
				Delirium Postoperative complications Institutionalisation rate	At adjusted logistic regression analysis, prehabilitation significantly reduced the incidence of delirium. No effect was observed for postoperative complications, institutionalisation and short-term mortality.	Positive Neutral Neutral
Tarazona-Santabalbina et al., 2019 [75]	310 patients ≥70 years Female: 63% Type of cancer: CRC (elective surgery) - 203 patients in intervention group (GS)	Retrospective	In GS group, the geriatrician performed a CGA and established a care plan, then applied and monitored by the geriatrician and multidisciplinary team. Control group was assessed daily by the General Surgery Service in accordance with the usual practice criteria.	LoS ICU admission Hospital readmissions	At baseline, patients in the GS group presented poorer clinical conditions than controls. LoS was similar in groups, but patients in the GS group stayed more frequently over ten days in hospital and were more frequently hospitalised and admitted to the ICU. No significant differences were observed between groups regarding readmissions and in-hospital and post-discharge mortality.	Neutral
				Delirium Geriatric syndromes Number of perioperative complications	54 patients experienced delirium (11.3% and 29.2% in GS and control group respectively, *p* < 0.001), and 49 patient experienced other geriatric syndromes (10.3%and 26.2% in the GS and control group respectively, *p* < 0.001). Serious complications were more frequent in the GS group (75.9% vs. 56.1%, *p* < 0.001).	Positive Positive Negative
Ommundsen et al., 2018 [64]	122 frail patients >65 years Mean age: 78.6 years Female: 52% Type of cancer: CRC (elective surgery) - Frailty is any of: (1) VES-13 > 2; (2) severe comorbidities; (3) cognitive impairment; (4) PP; (5) malnourishment - 53 patients in intervention arm	RCT	CGA followed by a tailored intervention or usual care.	LoS Reoperations Hospital readmissions	No differences in term of LoS between groups. No statistically significant differences between intervention and control group for reoperations (19% vs. 11%) or readmissions (16% vs. 6%).	Neutral
				Severity of postoperative complications	In the secondary analyses, a statistically significant difference in favour of the intervention in terms of lower CD grade I–V complications (*p* = 0.05) was found. No statistically significant differences between intervention and control group for CD grade II–V complications (68% vs. 75%) or 30-day survival (4% vs. 5%).	Positive
Shipway et al., 2018 [82]	682 cancer patients (84% were cancer patients) ≥60 years Mean age: 72 years Type of cancer: resectable CRC, esophagogastric cancer (undergoing surgery). - 132 patients in intervention group	Before-and-after	Preoperative CGA and corresponding interventions, postoperative patient co-management by a geriatrician. Geriatrician involvement also in the definition of the postoperative discharge plan and in the implementation of a rehabilitation program.	LoS	Intervention was associated with a LoS significant reduction (by 3.1 days) for all surgical patients aged > 60 years, with esteemed cost savings of approximately £300,000/annum. In patients admitted electively for GI surgery, LoS reductions did not reach statistical significance, although a trend reduction was seen indicating possibly greater reduction with advancing age.	Positive
				Medical complications	No statistically difference in term of medical complications (the reduction of LoS could reflect the prevalence of these).	Neutral
Souwer et al., 2018 [77]	86 patients ≥75 years Median age: 81 years Female: 51% Type of cancer: (stage I-III) CRC (elective surgery) - 86 patients in intervention cohort	Retrospective	Multidisciplinary pre- and rehabilitation (cohort 2014–2015): preoperative assessment with geriatric screening, subsequent CGA when indicated, rehabilitation care. Retrospectively identified historic control cohorts of patients operated at same centre (cohorts 2010–2011, 2012–2013).	LoS Readmission (within 30 days)	The number of patients with a prolonged LoS (>14 days) decreased from 27% in 2010–2011 to 13% in 2012–2013 and 6% in 2014–2015 (*p* = 0.001), with readmission rates of 3%, 8% and 8% respectively.	Positive Neutral
				Postoperative complications Adjuvant CT	Severe complications and cardiac complications after surgery were significantly reduced. Number of surgical and pulmonary complications did not differ between the three cohorts. Six patients in 2010–2011, seven in 2012–2013 and 11 in the study cohort received adjuvant CT.	Positive
Ho et al., 2017 [59]	74 patients >70 years Type of cancer: CRC (undergoing curative surgery)	RCT	All patients are randomised to either conventional surgical care or enhanced geriatric input.	LoS	The median LoS was statistically significantly shorter in the intervention group when compared to control (7.1 ± 4.0 days vs. 14.0 ± 10.9 days, *p* < 0.0001).	Positive
				Postoperative complications	Postoperative complications were significantly lower in the intervention group (16.2% vs. 54.1%, *p* < 0.001).	Positive
Indrakusuma et al., 2015 [79]	100 patients ≥70 years Type of cancer: CRC (elective resection). Cohort ISAR – (2008–2010): ISAR questionnaire was not used. Cohort ISAR + (2011–2013): ISAR questionnaire was used. Match-control comparison: 50 DOG patients are compared with 50 matched controls from cohort ISAR −.	Retrospective cohort and match-control study	Patients from cohort ISAR + with a positive ISAR score were referred to the geriatric specialists for DOG assessment. The assessment encompassed CGA and geriatric interventions (e.g., vitamin supplementation, dietary supplements, consult with cardiologist, transfusion, haloperidol prophylaxis).	LoS	LoS was only statistically significant shorter in the cohort (ISAR+ vs. ISAR-) comparison (but since 2011 the use of laparoscopic resection increased, preoperative workup improved, and a postoperative fast track program was implemented).	Neutral
				PoD Postoperative complications	Compared with controls, DOG patients were older and underwent laparoscopic resection more often. Hearing and cognitive impairment were more prevalent among DOG patients, as history of delirium. Even if significantly more at risk for postoperative complications, DOG patients had comparable postoperative outcomes as controls in general/surgical and medical complications. DOG patients had similar outcomes in mortality and PoD compared to controls.	Positive
Mak et al., 2014 [60]	78 patients >70 years Type of cancer: CRC (surgical treatment) - 79 patients in intervention arm - control group: registry (database of the same surgical department)	Prospective pilot study	Intervention group: perioperative assessment and active management of their pre-existing medical problems, nutritional status and social status were carried out. Patients were jointly managed perioperatively by the colorectal and geriatric teams with further input on discharge. Control group: standard care	LoS	The interventional group had shorter mean LoS (9.31 vs. 12.2 days; *p* < 0.0001).	Positive
				Discharge destination	Discharge destination (i.e., home, nursing home or rehabilitation hospital) in both groups was not different. Older patients who received geriatric input had lower 30-day morbidity when compared with controls (15.4% vs. 20.4%; *p* < 0.01).	Neutral
Hempenius et al., 2013 [63]	260 frail patients ≥65 years Mean age: 77 years Female: 73% Type of cancer: solid cancers (elective surgery) - 148 patients in the intervention group - randomisation stratified by cancer type - Frailty: GFI > 3	RCT	Geriatric liaison intervention to prevent PoD: pre-operative CGA by a geriatric team, individual treatment plan, daily visits by a geriatric nurse during the hospital stay and advice on emerging medical problems. The intervention focused on best supportive care and the prevention of delirium. Control group: standard care (additional geriatric care was only provided at the request of the treating physician).	LoS QoL	Median LoS was 8 days in both groups. No significant difference between the groups in most aspects of the SF-36 scale to estimate QoL, although intervention group did report significantly less bodily pain at discharge than at admission compared with the usual-care group (OR: 0.49, 95% CI: 0.29–0.82).	Neutral Positive
				Postoperative complications PoD incidence and severity Return to an independent preoperative living situation Care dependency	No significant difference between groups in number and type of complications. PoD occurred in 31 patients (11.9%). No significant difference in the incidence of PoD between the intervention and usual-care group as well as for severity of PoD. This was a significant difference in term of return to preoperative living situation and care in favour of the intervention group, as opposed to the care dependency.	Neutral Negative Positive

Abbreviations: CD: Clavien-Dindo; CGA: Comprehensive Geriatric Assessment; CRC: Colorectal Cancer; EQ-5D: EuroQol-5 Dimension; G8: Geriatric 8; GFI: Groningen Frailty Indicator; GS: CGA-based care; GI: Gastrointestinal; ICU: Intensive Care Unit; ISAR: Identification of Seniors At Risk (questionnaire); LoS: Length of Stay; MDT: Multidisciplinary team; OS: Overall Survival; RCT: Randomised Controlled Trial; PEERS: Programme for Enhanced Elderly Recovery; PERI-OP: Perioperative geriatric intervention; PoD: Postoperative Delirium; QoL: Quality of Life; 6-CIT: 6-Item Cognitive Impairment Test. White rows: cost-sensitive measures (measures of patient health conditions leading to cost-increasing or cost-decreasing effects); grey rows: measures of effectiveness of CGA intervention (with estimation of costs in two studies [81,86]).

**Table 3 cancers-14-03235-t003:** Studies conducted in medical settings, main features and results.

Authors	Participants	Type of Study	CGA and Geriatric Interventions	Effectiveness Measures Suitable for Cost-Effectiveness	Main Effect of CGA and Geriatric Interventions	Cost-Effectiveness Propensity
Lund et al., 2021 [67]	142 frail patients ≥70 years Median age: 75 years Female: 43% Type of cancer: stage II–IV CRC [receiving adjuvant (58%) or first-line palliative/downstaging CT (42%)] - Frailty: G8 ≤ 14 - 71 patients in intervention group	RCT	CGA-based intervention: CGA at the start of CT, follow-up after 2 months or more frequently if needed. Interventions included medication changes (62%), nutritional therapy (51%), physiotherapy (39%). Control group: standard care, co-existing health problems assessed by oncologist or GP.	Hospitalisations QoL	Hospitalisation during CT occurred with equal frequency in both groups. QoL (EORTC QLQ-C30, EORTC QLQ ELD-14) was better in intervention patients, with decreased burden of illness (*p* = 0.048) and improved mobility (*p* = 0.008).	Neutral Positive
				ADEs CT completion	Grade 3+ toxicity occurred in 39% of patients from the control arm and in 28% of patients from the CGA arm (*p* = 0.156). 20% in the intervention group and 30% in the control group discontinued CT due to toxicity (*p* = 0.173). More patients from the CGA arm completed the scheduled CT compared with controls (45% vs. 28%, *p* = 0.0366).	Neutral Positive
Mohile et al., 2021 [62]	718 frail patients ≥70 years Mean age: 77,2 years Female: 43% Type of cancer: incurable advanced (stage III-IV) solid cancer or lymphoma (starting a new treatment regimen with a high risk of toxic effects within 4 weeks) Frailty: at least one impaired CGA domain other than PP. - 349 patients in intervention arm	RCT	Intervention (CGA and CGA-guided management integrated into oncology care): patients completed CGA, oncologists received a tailored CGA summary and management recommendations. Usual care: patients completed CGA, no CGA summary or management recommendations were provided to oncologists (only alerts for significantly impaired scores on depression and cognitive screening were sent).	ADEs PP Falls	A lower proportion of patients in intervention group had grade 3–5 ADEs compared with usual care group (51% vs. 71%), with a reduced risk of ADEs (*p* = 0.0001). Patients in the intervention group had fewer falls (12% vs. 21%) with lower risk of having new falls (*p* = 0.0035) and had more medications discontinued (*p* = 0.015), reducing PP.	Positive
Choukroun et al., 2021 [85]	51 frail outpatients ≥75 years Mean age: 83,7 years Median age: 83 years Female: 57% Type of cancer: not-hematologic solid cancers [breast (27%), CRC (16%), metastatic cancer (42%)]. - patients with G8 score ≤14 - median number of chronic comorbidities (excluding primary cancer) was 4.0 [high blood pressure (69%), dyslipidaemia (29%), chronic renal failure (27%), heart failure (25%) and type 2 diabetes (24%)].	Prospective observational study	Five-stage process (outpatient): preparation phase; face-to-face pharmaceutical consultation with the patient; CGA performed by an ergotherapist and a geriatrician (geriatric tools; physical examination); pharmaceutical medication analysis (PP, PIM, DRP and DDI); final multidisciplinary medication review by the clinical pharmacist and a geriatrician team (recommendations to reduce DRP and optimise prescriptions). After the medication review, the proposals for prescription modification were sent to GPs and oncologists.	PIM ADE risk PP	A significant decrease was observed in prevalence of PIM use and ADE risk. A not significant trend was observed for a lower number of medications.	Positive
DuMontier et al., 2020 [65]	160 frail/prefrail patients [per-protocol intervention and control (*n* = 148)] ≥75 years Median age: 80,4 years Female: 45% Type of cancer: lymphoma, leukaemia, or multiple myeloma (transplant-ineligible patients). Randomization stratified by disease type. - 60 patients in intervention group - 48 patients had at least one visit with a geriatrician	RCT	Prefrail and frail patients were randomised to standard oncologic care or standard care plus consultation with a geriatrician. Geriatrician provided individualised interventions, if indicated, he communicated with patient’s primary care provider and utilized referral systems (e.g., psychiatry). Follow-up was encouraged, but not required. Most common interventions fell within the comorbidity/PP domain (81%); followed by nutrition (54%); function/falls (48%); cognition (31%) and depression/mood (17%).	LoS ED visits Unplanned hospitalisations	Consultation did not significantly reduce the incidence of ED visits, hospitalisations (6 months follow-up), or days in hospital.	Neutral
				Discussion on EoL goals	Consultation did improve the odds of having EoL goals of care discussions (OR = 3.12, 95% CI: 1.03–9.41).	Positive
Li et al., 2020 [61]	605 patients ≥65 years Mean age: 72 years Median age: 71 years Female: 59% Type of cancer: solid malignant neoplasm [GI (33.4%), breast (22.5%), lung (16%), GU (15%), gynaecologic (8.9%), other (4.1%)] (starting a new CT regimen). 71.4% of patients with stage IV cancer. - 402 patients in intervention arm	RCT	Before starting CT, both arms completed a baseline CGA and Fulmer SPICES assessment. In the intervention arm (GAIN), a geriatrics-trained multidisciplinary team (oncologist, nurse practitioner, social worker, physical/occupation therapist, nutritionist, and pharmacist) acted on CGA results and implemented interventions. In the control arm (SOC), CGA results were sent to treating oncologists for consideration without any input from the multidisciplinary team.	LoS ED visits Unplanned hospitalisations Hospital readmissions	No significant differences were observed in ED visits, unplanned hospitalisations, average LoS, unplanned hospital readmissions and OS between groups.	Neutral
				ADEs CT discontinuation	Compared to SOC, GAIN arm experienced a significant 10.1% reduction in the incidence of grade 3+ CT-related toxic effects. Reductions were observed for hematologic-only toxic effects (8.0% reduction) as well as non-hematologic-only toxic effects (8.2% reduction). No significant differences in CT dose modifications or discontinuations.	Positive Neutral
Nadaraja et al., 2020 [69]	96 patients ≥70 years Median age: 75.4 years Female: 47.9% Type of cancer: GU cancer (ovarian 32.3%, endometrial 8.3%, prostate 32.3%), bladder 12.5%, kidney 6.3%, NSCLC 8.3% (starting CT or targeted therapy for primary or recurrent disease). - 49 patients in intervention group	RCT	Control group: treatment decision based on oncologist’s clinical judgement. Intervention group if G8 ≤ 14: treatment decision based on G8 screening followed by CGA, a multidisciplinary team conference and interventions. Intervention group if G8 > 14: treatment decision based on oncologist’s clinical judgement.	Treatment completion ADEs	No impact on completion rate of planned oncologic treatment, but the intervention resulted in a borderline significant lower incidence of grade 3–4 toxicity.	Neutral Positive
Nipp et al., 2020 [70]	62 patients ≥65 years Median age: 72.3 years Female: 45% Type of cancer: incurable GI or lung cancer - 30 patients assigned to intervention group	RCT	Random assignment to usual care or intervention. Transdisciplinary intervention integrating geriatric and palliative care with oncology care (two visits with a geriatrician trained about geriatric-specific and palliative care issues and CGA).	QoL	Intervention patients presented less decrease in QoL decrement (FACT-G).	Positive
				Symptom burden Communication confidence	Intervention patients had reduced number of moderate/severe symptoms and improved confidence in communication compared to usual care.	Positive
Ørum et al, 2021 [71]	363 patients >70 years Median age: 75 years Female: 45% Type of cancer: head and neck, (4%), lung (47%), upper GI (23%), CRC (26%). - 152 patients in intervention arm	RCT	All patients received CGA at baseline performed by a multidisciplinary team with evaluation of patient health status. Intervention group received a tailored follow-up by a multidisciplinary team. Follow-up lasted 90 days, performed in-hospital (either in the outpatient clinic or during hospitalisation), in the patient own home, or as phone calls.	Hospitalisations	No significant impact on hospitalisation (47% of intervention vs. 55% of controls).	Neutral
				Completion of planned treatment ADLs Physical Performance	No differences in ability to complete the treatment, ADLs or physical performance were found.	Neutral
Soo et al., 2020 [58,87]	130 patients >70 years Type of cancer: solid organ cancer or diffuse large B-cell lymphoma [candidates for systemic anticancer therapy (CT, targeted therapy or immunotherapy)].	RCT	Intervention group received integrated oncogeriatric care: CGA and management integrated with standard oncology care. The group was co-managed by a geriatrician during oncological treatment, undergoing a CGA, standardised personalised interventions and receiving referrals, supportive care information, encouragement of physical activity, management of comorbidities, medication reconciliation and advance care planning. Control group: managed by oncologist only, without input from geriatrician.	QoL Unplanned hospital admissions ED visits	Significant differences favouring the intervention group over the usual care group were seen in QoL—assessed using EORTC QLQ-C30 and EORTC QLQ-ELD14 at 0, 12, 18 and 24 weeks—and unplanned hospital admissions (−1.2 admissions person-years in intervention group; 41% less) and ED visits (39% less). Intervention group presented significantly better ELFI than usual care group at all follow-ups.	Positive
				Treatment discontinuation	Significant differences favouring the intervention group over the usual care group were seen in early treatment discontinuation (32.9% vs. 53.2%, respectively).	Positive
Ramsdale et al., 2018 [84]	40 patients ≥70 years Mean and median age: 77 years Female: 45% Type of cancer: advanced solid cancer or lymphoma. Impairment in at least one geriatric domain other than PP. - 20 patients in intervention arm	Descriptive comparison study (subset of patients enrolled in RCT)	All patients received CGA at baseline, prior to starting antineoplastic therapy. In the intervention group, oncologists were given results of CGA. In the control group, they received no information.	Addressing PIM and PP	Physician-initiated discussions were higher in intervention (73% vs. 49%, *p* = 0.006). More PP concerns brought up per patient (4.1 vs. 2.6, *p* = 0.07) and “addressed” in intervention compared with control (59% vs. 45%, *p* = 0.1). Medication management concerns were addressed more commonly in intervention (79% vs. 38%, *p* = 0.003). Supportive care medication concerns were more often addressed in control group (58% vs. 18%, *p* = 0.008).	Positive
Magnuson et al., 2018 [83]	71 patients ≥70 years Mean age: 76 years Female: 44 % Type of cancer: advanced (stages III or IV) solid tumour malignancy. - 37 patients in intervention group [recommendations for GA management interventions were relayed to the primary oncologist within the target time-frame in 34 patients (92%)].	Prospective randomised pilot study	In intervention arm an algorithm was used to guide GA management recommendations. The coordinator scored the GA and identified impairments, then summarised GA impairments with management recommendations and delivered recommendations to the patient’s primary oncologist within 1 week of assessment. At the 3-month follow-up timepoint, the primary oncologist reported whether these recommendations had been implemented.	Hospitalisations	Prevalence of hospitalisation did not differ between the two groups.	Neutral
				ADEs Treatment continuity	Incidence of grade 3–5 CT toxicity did not differ between the two groups. Dose reductions, dose delays, and early treatment discontinuation also did not differ between the two groups.	Neutral
Puts et al., 2018 [73] and Sattar et al., 2019 [88]	61 patients ≥70 years Mean age: 75 years Female: 36% Type of cancer: (stage I–IV) GI, GU, or breast cancer. - 31 patients allocated to intervention arm	RCT	Intervention: CGA, interventions, first follow up. 3 months after CGA, follow-up appointment if needed. Treating physician received the summary of findings and interventions that would be implemented by the clinical intervention team. Control group: usual care from the oncology team.	QoL ED visits Family physician visit	Slight benefit in QoL for intervention patients, but results at all timepoints presented no statistically significant difference. No significant differences in number of ED and GP visits, even if slightly lower rates in intervention group.	Neutral
				IADL impairment	Fewer patients with IADL impairment ≥ 1 in intervention than in control group at baseline. At six months, the proportion of those with ≥ 1 IADL impairment was similar.	Neutral
Corre et al, 2016 [66]	494 patients ≥70 years Median age: 77 years Female: 26% Type of cancer: advanced (stage IV) NSCLC (starting CT). Patients with PS of 0 to 2. - 243 patients in the intervention arm	RCT	All patients had a CGA performed by their regular cancer physician. Intervention group: experimental CGA-based allocation of the same CT or BSC. Control group: standard strategy of treatment allocation (based on PS and age).	QoL QoL-adjusted survival	Although QoL utility scores at baseline were not different between the arms, they always were higher (although not significantly) in the CGA arm than in the standard arm at each subsequent evaluation, with no evident negative impact of the 23% of patients who received exclusive BSC. The difference in QoL utility scores was significant only at week 36 (*p* = 0.02).	Neutral
				ADEs CT failures for toxicity	Treatment objective response rate and disease control rate in CGA arm and control showed no difference. Intervention had significantly less all grade toxicity than control (85.6% vs. 93.4%, *p* = 0.015) and fewer treatment failures as result of toxicity (4.8% vs. 11.8%, *p* = 0.007). Percentage of patients with all grade ADEs was significantly higher in control than intervention (93.4% vs. 85.6%, *p* = 0.015), but not significantly for grade III-IV (71.3% vs. 67.9%, *p* = 0.41).	Positive
Kalsi et al., 2015 [74]	135 patients ≥70 years Mean age: 75 years Female: 45% Type of cancer: GI (55%), other (45%) (starting CT with or without RT). - 65 patients in intervention group (2011–2013) - 70 patients in observational control group (2010–2012).	Prospective cohort comparison study	The intervention group underwent risk stratification using a patient-completed screening questionnaire and high-risk patients received geriatrician-delivered CGA. The observational control group received standard oncology care.	CT outcome ADEs CT completion	Geriatrician-delivered CGA was associated with better outcomes. No significant trend for a lower grade 3+ toxicity rate in intervention (43.8% 3+ toxicity rate in the intervention group and 52.9% in the control). More participants in intervention completed treatment as planned (*p* = 0.006) and fewer required treatment modifications (*p* = 0.006).	Positive Neutral Positive
Bourdel-Marchasson et al., 2014 [68], Regueme et al., 2021 [89]	336 patients ≥70 years Mean age: 78 years Female: 49% Type of cancer: carcinomas (colon, stomach, pancreas and biliary tract, ovary, prostate, bladder, and lung) and lymphomas (treated with CT). Patients with at least KPS ≥ 50% and at risk of malnutrition (17 ≤ MNA ≤ 23.5).	RCT	The usual care received usual dietary recommendations. The intervention group received usual care and nutritional intervention. Counselling was based on face-to face interviewing and dietary advice cards and involved caregivers or relatives if possible.	Hospitalisation QoL	Analyses were performed on an ITT basis. The intervention was no beneficial for hospitalisation (intervention presented not significantly lower hospitalisations) and 1 and 2-year mortality (similar in both groups). Cancer cachexia anti-anabolism may explain this lack of effect. The intervention did not modify the HRQoL changes in comparison with routine care.	Neutral
				ADEs CT management Dietary intake	Diet counselling was efficient in increasing dietary intake but had no beneficial effect on CT management (dosage, changes, arrest). There were more usual care patients with grade 3 to 4 infections than in intervention group, but the robustness analysis did not confirm the difference in the incidence of severe infections.	Neutral
Rao et al., 2005 [86]	99 frail patients ≥65 years Mean age: 74 years Female: 2% Type of cancer: prostate, lung, hematologic, GI, head/neck, bladder, renal cancer, ill-defined malignancies. Frailty is at least 2 among: dependence in ADL, stroke/unplanned admission (last 3 months), previous falls, critical ambulation, malnutrition, dementia, depression, prolonged bed rest, incontinence.	Secondary analysis (of RCT)	Hospitalised on a medical or surgical ward, after stabilisation of acute illness, randomised to receive care in a geriatric inpatient unit/geriatric outpatient clinic/both/neither. In geriatric evaluation and management units (inpatient and outpatient geriatric units) core teams provided CGA and patient management.	LoS Direct costs for care QoL	Geriatric evaluation and management inpatient units impacts the QoL (SF-36) in the management of bodily pain and mental and emotional health (no difference in SF-36 general scores between groups). These effects were achieved with no overall increase in hospitalisation or cost of care over the year of the study. No significant differences in LoS. No effect on mortality.	Neutral Positive Positive

Abbreviations: ADLs: Activities of Daily Living; ADEs: Adverse Drug Events; AEs: Adverse Events; BSC: Best Supportive Care; CT: Chemotherapy; CGA: Comprehensive Geriatric Assessment; CRC: Colorectal cancer; DDI: Drug–Drug Interaction; DRP: Drug Related Problems; ED: Emergency Department; ELFI: Elderly Functional Index; EoL: End of Life; EQ-5D: EORTC: European Organisation for Research and Treatment of Cancer; EORTC QLQ-C30: EORTC Core Quality of Life Questionnaire; EORTC QLQ-ELD14: EORTC Quality of Life Questionnaire—Elderly Cancer Patients Module; EuroQoL-5 Dimension; EQ-5D-3L: 3-level version of EQ-5D; FACT-G: Functional Assessment of Cancer Therapy-General; GPs: General Practitioners; GU: genitourinary; HRQoL: Health-Related Quality of Life; IADLs: Instrumental Activities of Daily Living; ITT: Intention-To-Treat; KPS: Karnofsky PS; LoS: Length of Stay; MNA: Mini Nutritional Assessment; NSCLC: Non-Small-Cell Lung Cancer; OS: Overall Survival; PFS: Progression Free Survival; PIM: Potentially Inappropriate Medication; PP: polypharmacy; PS: Performance Status; RCT: Randomised Control Trial; RT: Radiotherapy; SF-36: Short Form-36 Health Survey; SPICES: Sleep disorders—Problems with eating/feeding–Incontinence–Confusion–Evidence of Falls–Skin Breakdown; TFFS: Treatment Failure-Free Survival; VES-13: Vulnerable Elders Survey. White rows: cost-sensitive measures (measures of patient health conditions leading to cost-increasing or cost-decreasing effects); grey rows: measures of effectiveness of CGA intervention (with the estimation of costs in two studies [81,86]).

**Table 4 cancers-14-03235-t004:** Most recurrent reported measures for effectiveness of CGA interventions.

Measures	Evidence in Identified Studies								
**LoS**	Nipp et al., 2022 [72] RCT (98 patients)	Koh et al., 2021 [81] Before-and-after (81 patients)	DuMontier et al., 2020 [65] RCT (160 patients)	Li et al., 2020 [61] RCT (605 patients)	Shahrokni et al., 2020 [76] Retrospective (1892 patients)	van der Vlies et al., 2020 [78] Retrospective (433 patients)	Janssen et al., 2019 [80] Before-and-after (627 patients)	Tarazona-Santabalbina et al., 2019 [75] Retrospective (310 patients)	

	positive	positive	neutral	neutral	negative	neutral ***	neutral	neutral ***	
	Ommundsen et al., 2018 [64] RCT (122 patients)	Shipway et al., 2018 [82] Before-and-after (682 patients)	Souwer et al., 2018 [77] Retrospective (86 patients)	Ho et al., 2017 [59] RCT (74 patients)	Indrakusuma et al., 2015 [79] Retrospective cohort study (100 patients)	Mak et al., 2014 [60] Prospective (78 patients)	Hempenius et al., 2013 [63] RCT (260 patients)	Rao et al.2005 [86] Secondary analysis (of RCT) (99 patients)	

	neutral	positive	positive	positive	neutral	positive	neutral ***	neutral	
**ADEs**	Lund et al., 2021 [67] RCT (412 patients)	Mohile et al., 2021 [62] RCT (718 patients)	Choukroun et al., 2021 [85] Prospective (51 patients)	Li et al., 2020 [61] RCT (605 patients)	Nadaraja et al., 2020 [69] RCT (96 patients)	Magnuson et al., 2018 [83] Prospective (71 patients)	Corre et al., 2016 [66] RCT (492 patients)	Kalsi et al., 2015 [74] Prospective (135 patients)	Bourdel-Marchasson et al., 2014 [68] RCT (336 patients)

	positive	positive	positive	positive	positive	neutral	positive	neutral	neutral

* Frail patients in the intervention group. Abbreviations: ADEs: Adverse Drug Events; LoS: Length of Stay; RCT: Randomised Control Trial.

**Table 5 cancers-14-03235-t005:** Costs for hospitalisation and adverse events treatments.

Country	Cost Estimates
United States [98]	Median cost per day of inpatient visits for older patients with cancer: USD 2108—USD 3468
United States [97]	Direct cost of neutropenia per episode: USD 1893 (outpatient)/USD 2893 (inpatient)—USD 38,583 (febrile neutropenia hospitalisation)
UK and Europe [97]	Direct cost of neutropenia per episode: USD 300 (non-febrile cases)—USD 32,395 (older breast cancer patients)
United States [97]	Direct cost of thrombocytopenia per cycle/episode: USD 1035—USD 5328
Europe [97]	Direct cost of thrombocytopenia per cycle/episode: USD 790—USD 2523
United States [97]	Direct cost attributable of anaemia per year: USD 18,418—USD 69,478
Canada and Europe [97]	Total cost of anaemia per episode: USD 124—USD 2704
United States and Europe [99]	Average total cost per ICU/day: estimated at EUR 1200
Europe [99]	Direct costs per sepsis patient (ICU): estimates of EUR 23,000—EUR 29,000.
United States [99]	Direct costs per sepsis patient (ICU): estimates of EUR 34,000
United States [100]	Costs for severe sepsis cancer hospitalisation: USD 27,400 Costs for surgical cancer severe sepsis hospitalisation: USD 48,000 Costs for medical cancer severe sepsis hospitalisation: USD 18,200 Costs for non-severe sepsis cancer hospitalisation: USD 8700
United States [101]	Healthcare costs per delirious patient per year: USD 60,516—USD 64,421

## Data Availability

Not applicable.

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
