# Peer review of "Exploring Cost-Effectiveness of the Comprehensive Geriatric Assessment in Geriatric Oncology: A Narrative Review"

_cancers, 2022, doi:10.3390/cancers14133235_

Round 1

Reviewer 1 Report

The authors have produced a comprehensive, well-written narrative review on the available evidence of utility of a CGA taking into account cost-effectiveness in older cancer patients. Though there are very few studies on the topic, most of which of poor quality, this issue is a topic of major relevance and as the authors clearly state, should be matter of dedicated research.

I have some minor suggestions that author should address:

-       The authors should state in the methods section the definition of cost-effectiveness propensity, given a qualitative assessment (positive Vs negative Vs neutral) is provided in Table 1 and 2.

-       The authors should explain their definition of “intermediate” and “final” outcomes of CGA-based interventions, since they describe separately “Studies reporting intermediate and final outcomes of CGA-interventions” and “Studies reporting intermediate outcomes of CGA: treatment toxicity and other complications”. Indeed it is not clear why chemotherapy toxicity is described as an intermediate outcome when toxicity – especially when leading to treatment discontinuation – can be a primary endpoint itself or as a combined endpoint in oncology trials.

-       Line 37 and throughout the manuscript: please omit if possible the term elderly (cfr. Lundebjerg NE,  JAGS 2017, 65:1386–1388) preferring the term “older”.

-       Table 3 and line 555: Evidence is uncountable, I would not use the plural “evidences”.

-       Line 464: after full stop, numbers should be written in letters

Author Response

Dear Reviewer, 

    We are pleased to submit the revision on the Manuscript titled “Exploring cost-effectiveness of the Comprehensive Geriatric Assessment in geriatric oncology: a narrative review”.

We thank you for the comments and suggestions on our Manuscript. As a result, we have revised the original version taking into account your comments. The thorough review and valuable suggestions have helped us improve the Manuscript. We have uploaded the revised Manuscript using track changes.

If for any reason the expectations have not been met, we are readily available to respond further and welcome the opportunity to do so.

Thank you for receiving our revisions. We appreciate your time and look forward to your response.

Sincerely

Reviewer 2 Report

Dear authors, 

This review aims to assess the added value of CGA in the management of clinical outcomes and cost-effectiveness among older cancer patients 

Title: the title should be modified by removing the term "narrative review" in favor of "systemic review"

INTRODUCTION

the intro is a bit too long

please consider a shorter intro 

METHODS

Overall, the methods reported are those of a systematic review. This should be clearly specify by adding the PRISMA statement with reference 

RESULTS

It could be interesting to summarize the 29 studies included in this review with a few sentences: for example, the mean age of the patients included varied between X and Y years, there were X RCT; Y observational studies; cancer sites considered...

Please add reference alongside the text when some studies are reported and whenever necessary +++

DISCUSSION

Instead of starting the discussion with "summarizing this review is difficult", the authors should rather insist on the main result which is the few studies identified on the subject +++

then the discussion should focus on the reasons for this result according to the authors 

what are the real intentions of having such results in the literature?

the authors should develop this axis more because as it stands the discussion is too long and we get lost in reading

Author Response

(The authors gave the same response as above.)
